# An Atomistic-Based Nonlinear Plate Theory for Hexagonal Boron Nitride

**DOI:** 10.3390/nano11113113

**Published:** 2021-11-18

**Authors:** Kun Huang, Jiye Wu, Yajun Yin

**Affiliations:** 1Department of Engineering Mechanics, Faculty of Civil Engineering and Mechanics, Kunming University of Science and Technology, Kunming 650500, China; 2School of Physical and Mathematical Sciences, Nanjing University of Technology, Nanjing 211816, China; wujiye@njtech.edu.cn; 3Department of Engineering Mechanics, School of Aerospace, Tsinghua University, Beijing 100084, China; yinyj@tsinghua.edu.cn

**Keywords:** hexagonal boron nitride, graphene, DREIDING force field, Föppl-von Karman plate theory, Gaussian curvature, Yakobson paradox

## Abstract

Through the continuity of the DREIDING force field, we propose, for the first time, the finite-deformation plate theory for the single-layer hexagonal boron nitride (h-BN) to clarify the atomic source of the structure against deformations. Divergent from the classical Föppl-von Karman plate theory, our new theory shows that h-BN’s two in-plane mechanical parameters are independent of two out-of-plane mechanical parameters. The new theory reveals the relationships between the h-BN’s elastic rigidities and the atomic force field: (1) two in-plane elastic rigidities come from the bond stretching and the bond angle bending; (2) the bending rigidity comes from the inversion angle and the dihedral angle torsion; (3) the Gaussian rigidity only comes from the dihedral angle torsion. Mechanical parameters obtained by our theory align with atomic calculations. The new theory proves that two four-body terms in the DREIDING force field are necessary to model the h-BN’s mechanical properties. Overall, our theory establishes a foundation to apply the classical plate theory on the h-BN, and the approach in this paper is heuristic in modelling the mechanical properties of the other two-dimensional nanostructures.

## 1. Introduction

Hexagonal boron nitride (h-BN) and graphene have the same lattice structure, excellent mechanical and electrical properties, and many potential applications [1,2,3,4,5]. Compared with graphene, h-BN’s mechanical property research is still not intensive [6,7]. The mechanical properties of graphene and carbon nanotubes have been researched for around thirty years. However, two critical mechanical problems have not yet been thoroughly clarified [1,8,9,10,11,12,13,14]. First, how does graphene resist out-of-plane deformations? Second, is the classical elasticity theory still valid for the one-atom-thick two-dimensional (2D) nanostructures? In the classical elasticity theory, an initial flat 2D structure can model as the film or the plate. The differences between the two structures are as follows [15,16]: the film can only resist in-plane tension and shear, while the plate can resist in-plane deformations and out-of-plane bending and twisting. Atomic calculations show that one-atom-thick graphene and h-BNs can resist in-plane and out-of-plane deformations [1,7,12,14]. Therefore, it is natural to use the classic plate theory to describe their mechanical properties. Based on the above assumption, many theories have been developed [1,7]. However, applying the plate theory to single-layer atomic structures may induce two problems. First, the macroscopic plate bending resistance comes from the opposite tension and compression condition on both sides of the midplane [16]. For one-atom-thick graphene or h-BNs, it is impossible for an atom to be in both tension and compression. Second, the graphene’s bending resistance also induces another widely mentioned argument, namely, the so-called Yakobson paradox: the uncertainty of the single-layer atomic structure’s thickness leads to the uncertainty of the bending mechanical properties (corresponds to the uncertainty of the bending rigidity) [1,9]. Overall, the two critical problems mentioned above cause many controversies when the classic plate theory is applied to one-atom-thick nanostructures.

In the classical elasticity theory, the deformation energy density of the Föppl-von Karman plate is [15,16]
(1)$$F_{p}=\frac{1}{2}k_{b}^{p}{\left(2J\right)}^{2}-k_{g}^{p}Q+\frac{1}{2}k_{B}^{p}{\left(2H\right)}^{2}+k_{G}^{p}G$$

Here $2J=\left(\varepsilon_{xx}+\varepsilon_{yy}\right)$, $Q=\left(\varepsilon_{xx}\varepsilon_{yy}-\varepsilon_{xy}^{2}\right)$ are the invariants of the 2D strain $\varepsilon_{ij}$ in the rectangular coordinate system (O;x,y). Lets the plate midplane displacement be w(x,y) in the z direction, then $H\approx \left(\partial^{2}w/\partial x^{2}+\partial^{2}w/\partial y^{2}\right)/2$ and $K\approx \left(\partial^{2}w/\partial x^{2}\right)\left(\partial^{2}w/\partial y^{2}\right)-{\left(\partial^{2}w/\partial x\partial y\right)}^{2}$ are the average curvature and the Gaussian curvature of the midplane. In Equation (1), $k_{b}^{p}$ and $k_{g}^{p}$ are the in-plane rigidity’s parameters, $k_{B}^{p}$ and $k_{G}^{p}$ are the out-of-plane bending rigidity and the Gaussian curvature rigidity (the torsional rigidity). The rigidity coefficients in Equation (1) are the functions of the elastic constants and the plate thickness [15,16]:(2)$$k_{b}^{p}=\frac{Eh}{1-\nu^{2}},\;k_{g}^{p}=\frac{Eh}{1+\nu },\;k_{B}^{p}=\frac{Eh^{3}}{12\left(1-\nu^{2}\right)},\;k_{G}^{p}=\frac{-Eh^{3}}{12\left(1+\nu \right)}$$

Here, E and ν are the three-dimensional (3D) Young’s modulus and the Poisson’s ratio, and h is the plate thickness. According to the classical macroscopic plate theory, has
(3)$$k_{B}^{p}/k_{b}^{p}=-k_{G}^{p}/k_{g}^{p}=Eh^{2}/12$$

Equation (2) indicates that the out-of-plane bending and torsional rigidities are related to the plate’s thickness. In detail, a thicker plate has a more robust ability to resist the out-of-plane deformations, and a plate with zero thickness cannot resist out-of-plane deformations. Since one-atom-thick graphene and h-BNs cannot define an explicit thickness, the Yakobson paradox is induced. Another critical question is whether the Gaussian curvature affects the mechanical properties of h-BNs and graphene [11,17,18,19] (this corresponds to whether the two materials have out-of-plane twisting resistance). For graphene, current researchers have presented contradictory results on this issue. Wu [17] and Atalaya [18] showed that the graphene’s deformation energy does not include the Gaussian curvature term through the continuity of the first-generation Brenner atomic potential and valence bond force field. Hence the two theories mean that graphene has not the out-of-plane twisting resistance. Nonetheless, Tu [19] and Davini [20] suggested that graphene has the negative Gauss curvature rigidity by continuity of the Lenosky force field and the 2nd-generation Brenner potential. The density functional tight-binding (DFTB) calculations also showed that graphene has a negative Gaussian curvature rigidity [1,21]. To solve this inconsistency, researchers have put in a great deal of effort. Recently, Huang and co-workers obtained a modified Föppl-von Karman plate theory for graphene using the tight-binding (TB) theory of covalent bond [11]. For the first time, this theory gives the relationships between all graphene’s mechanical parameters and chemical bonds under finite deformations. In detail, the rigidities against the in-plane extension and shear, which associates with 2D Young’s modulus and Poisson’s ratio, come from bond length changes and the misalignments of the angular of σ-orbitals after deformations. The bending resistance, which associates with the mean curvature rigidity, comes from the off-plane pyramidalization of the three adjacent π-orbitals and the twist of the adjacent π-orbitals; and the out-of-plane twisting rigidity, the Gaussian rigidity, only connects with the twist of the adjacent π-orbitals. Huang’s theory provides the physical explanation of the Yakobson paradox and theoretically demonstrates that graphene can be modeled as a modified elasticity plate. This theory also provides an instructive example to model the other 2D nanomaterials’ mechanical properties.

Existing experiments and atomic calculations show that the in-plane Young’s modulus and bending rigidity of h-NB are slightly lower than graphene [1]. The h-BN’s chemical bonds are similar to graphene: the hexagonal symmetry structure is constructed by the $sp^{2}$-hybridized σ-bonds between nitrogen and boron, and the remaining p-orbitals form π-bonds. Therefore, a straightforward deduction is that h-BN and graphene have a similar mechanical model. Song [22] established a continuous model (CM) of h-BN nanotubes by the Brenner potential. This theory shows that h-BN has no Gaussian curvature term and cannot resist torsion. Through the UFF atom potential, Genoese obtained the h-BN’s in-plane Young’s modulus and Poisson’s ratio that are consistent with the atomic calculations [23]. However, it is confusing that Genoese’s theory leads to a negative out-of-plane Poisson’s ratio if the model’s bending and the Gaussian stiffnesses are compared with the classical plate theory. Current research indicates that the h-BN’s mechanical parameters are closely related to the used force fields. For example, Rajan has developed a multi-body force field to consider the polar nature of the B-N bond [24]. Molecular dynamics (MD) simulations show that in-plane mechanical parameters obtained by the Rajan’s force field are very close to the results of experiments and DFTB calculations. However, based on the Rajan’s force field, the out-of-plane bending rigidity is 2.13 eV. That is much greater than 1.34 eV obtained by DFTB calculations [1,14]. It is necessary to use a simple atomic force field of h-BNs that can capture the primary deformation energy and the explicit physical mechanism. The TB theory has demonstrated that the Föppl-von Karman plate can describe the graphene’s mechanical properties with four independent parameters [11]. However, modelling the h-BN’s mechanical properties is still an open question. In this paper, we will use the DREIGING force field, which is widely used in the MD’s simulations of 2D nanomaterials, to validate that h-BN can model as the modified Föppl-von Karman plate. Our research may help clarify the long-standing controversies over the mechanical properties of h-BN and graphene. The approach in this paper can be a reference for research concerning other 2D nanomaterials.

## 2. Method

If the B-N bond’s heteropolarity is removed, h-BN and graphene have identical lattice structures and similar chemical bonds [25]. Hence an intuitive hypothesis is that they have similar force fields. Huang and co-workers have obtained the TB force field of graphene that includes bond stretching, the bond angle bending, the angle of off-plane pyramidalization, and the two adjacent π-orbitals twisting angle [11]. The off-plane pyramidalization angle and the inversion angle represent the four adjacent atoms’ deviation from the initial plane. Under the first-order approximation, we will illustrate later that the dihedral angle change is equivalent to the π-orbital twisting angle of adjacent two atoms. Therefore, the atomic force field of h-BNs should include the bond stretching, bond bending, inversion angle, and dihedral angle because h-BN’s force field is similar to graphene. The above four items can be captured by the DREIDING force field without nonbonding effect [26]: $U=U_{l}+U_{\theta }+U_{\chi }+U_{\omega }$, in which the four components represent respectively the potential energy induced by the bond stretching, bond angle bend, inversion angle and dihedral angle, as shown in Figure 1. By means of MD calculations with the modified parameters of the DREIDING force field, the in-plane elastic parameters and the bending stiffness are obtained. They are in good agreement with the DFTB calculations [27]. The present paper will utilize the DREIDING force field to study the h-BN’s mechanical properties. The potential energy expression of the DREIDING force field is [26]:(4)$$\begin{array}{l} {\bar{U}}_{bonded}=\frac{k_{l}}{2}\sum {\left(l_{i}-l_{0}\right)}^{2}+\frac{k_{\theta }}{2}\sum {\left(\theta_{i}-\theta_{0}\right)}^{2}\\ +\frac{k_{\chi }}{3}\sum \left(1-\cos\chi_{i}\right)+\frac{k_{\omega }}{8}\sum \left(1-\cos2\omega_{ijkl}\right) \end{array}$$

Here $l_{i}$, $\theta_{i}$, $\chi_{i}$ and $\omega_{i}$ are the deformed bond length, bond angle, inversion angle, and dihedral angle. The summation ∑ is taken over all lengths or angles. $\theta_{0}=2\pi /3$ and $l_{0}=0.145\;nm$ is the initial bond angle and bond length, as shown in Figure 1. Ignoring the difference between boron and nitrogen atoms, the potential energy of one atom, for example the boron atom at the point *O*, rewrite as
(5)$$\begin{array}{l} U_{bonded}=\frac{k_{l}}{4}\sum_{i=1}^{3}{\left(l_{i}-l_{0}\right)}^{2}+\frac{k_{\theta }}{2}\sum_{i=1}^{3}{\left(\theta_{i}-\theta_{0}\right)}^{2}\\ +\frac{k_{\chi }}{3}\sum_{i=1}^{3}\left(1-\cos\chi_{i}\right)+\frac{k_{\omega }}{16}[\sum_{i=5,6;j=2,3}\left(1-\cos2\omega_{i1Oj}\right)\\ +\sum_{i=8,9;j=1,3}\left(1-\cos2\omega_{i2Oj}\right)+\sum_{i=4,5;j=1,2}\left(1-\cos2\omega_{i3Oj}\right)] \end{array}$$

Different researchers adopted different values in Equation (5). Table 1 lists several widely used data sets of h-BN and graphene (in DREIGING, the graphene’s bond order is 1.5 [28], and the h-BN’s bond order is 1.09 [29]).

Within the existing research, the bond stretching and bond angle bending terms were adopted by most researchers. However, how to decide the inversion angle and the dihedral angle terms is a controversial question. For example, Li developed a structural mechanics approach to study the carbon nanotube’s mechanical properties by using bond stretching, bond bending, and dihedral angle terms [30]. This method has been widely applied to study the mechanical properties of the h-BN and graphene [31,32,33]. However, another opposite pathway uses only the inversion angle term to represent the change in the off-plane potential energy [18,24]. We will show later that both the inversion angles and the dihedral angles are necessary, and their absence may induce incorrect results.

To obtain the continuum limit form of the h-BN’s deformation energy, we make two geometric presumptions: first, all atoms in the undeformed plane distribute on the continuous deformed curved surface S, as shown in Figure 1; second, we ignore the impact of the in-plane deformations on the out-of-plane deformations. The second presumption is necessary to derive the Föppl-von Karman plate theory from the classical nonlinear elastic theory [15]. In the present research, it is not necessary to distinguish boron and nitrogen atoms. Thus, only the central boron atom, the three first-neighboring nitrogen atoms, and the six second-neighboring boron atoms are considered, as shown in Figure 1. 

Here we have
(6)$$\begin{array}{l} {\left(\theta_{i}-\theta_{0}\right)}^{2}\approx \frac{4}{3}{\left(\cos\theta_{i}+\frac{1}{2}\right)}^{2},\\ 1-\cos\chi_{k}=1-\left|n_{iOj}\times d_{Ok}\right|,\\ 1-\cos2\omega_{ijkl}=2\left(1-\cos^{2}\omega_{ijkl}\right)=2\left[1-{\left(n_{iOj}\cdot n_{Ojk}\right)}^{2}\right] \end{array}$$

In Equation (6), the bold letters $n_{pqr}$ represents the unit vector perpendicular to the plane formed by the three adjacent atoms p, q and r; $d_{Ok}$ is the unit vector of $r_{Ok}$. We decompose $n_{pqr}$ into two components at point q: $n_{pqr}^{\tau }$ in the tangent plane of surface S, and $n_{pqr}^{n}$ in the normal direction of the surface S. Due to $\left|n_{pqr}^{\tau }\right|\ll \left|n_{pqr}^{n}\right|$ under the small curvature, we have $n_{iOj}\approx n_{O}$ and $n_{Ojk}\approx n_{j}$. Here $n_{O}$ and $n_{j}$ represent the unit normal vectors at O and j positions. Considering $1-{\left(n_{O}\cdot n_{j}\right)}^{2}\approx 2\left(1-n_{O}\cdot n_{j}\right)$, Equation (5) is rewritten as
(7)$$\begin{array}{l} U_{bonded}=\frac{k_{l}}{4}\sum_{i=1}^{3}{\left(l_{Oi}-l_{0}\right)}^{2}+\frac{2k_{\theta }}{3}\sum_{i=1}^{3}{\left(\cos\theta_{iOj}+\frac{1}{2}\right)}^{2}\\ +\frac{k_{\chi }}{3}\left(3-\left|n_{O}\times d_{O1}\right|-\left|n_{O}\times d_{O2}\right|-\left|n_{O}\times d_{O3}\right|\right)\\ +\frac{k_{\omega }}{2}\sum_{i=1,2,3}\left(1-n_{O}\cdot n_{i}\right) \end{array}$$

Considering the finite displacement theory [34], has
(8)$$l_{Oi}-l_{0}=\left(d_{Oi}^{0}\cdot \varepsilon_{ij}\cdot d_{Oi}^{0}\right)l_{0},$$
(9)$$\cos\theta_{iOj}=d_{Oi}^{0}\cdot d_{Oj}^{0}+2d_{Oi}^{0}\cdot \varepsilon_{ij}\cdot d_{Oj}^{0}$$

Here $d_{Oi}^{0}$ and $d_{Oj}^{0}$ denote the unit vector of the undeformed $r_{Oi}^{0}$ and $r_{Oj}^{0}$ correspondingly. Substituting Equations (8) and (9) into the first two terms of Equation (5), and through complicated but straightforward calculations, gets (calculation details refer to Refs. [11,17,22])
(10)$$\begin{array}{l} \sum_{i=1}^{3}{\left(l_{Oi}-l_{0}\right)}^{2}=l_{0}^{2}\left[\frac{9}{8}{\left(2J\right)}^{2}-\frac{3}{2}Q\right],\\ \sum_{i=1}^{3}{\left(\frac{1}{2}+\cos\theta_{iOj}\right)}^{2}=\left[\frac{9}{4}{\left(2J\right)}^{2}-6Q\right] \end{array}$$

In order to obtain the continuous limit of the two off-plane four-body terms, we approximate the deformed h-BN’s surface as the Monge patch [15,35]. We supposes that the atom O stays in the initial position during deformation, and let $n_{Ox}=r_{x}\left(0,0\right)/\left|r_{Ox}\right|$, and $n_{Oy}=r_{y}\left(0,0\right)/\left|r_{Oy}\right|$, here $r_{x}\left(0,0\right)=\frac{\partial r}{\partial x}\left(0,0\right)$, and $r_{y}\left(0,0\right)=\frac{\partial r}{\partial y}\left(0,0\right)$. According to the classic finite deformation plate theory [15], approximately gets $n_{Ox}⊥n_{Oy}$, $n_{Ox}$ and $n_{Oy}$ are parallel to x and y axis correspondingly. Under the coordinate system $\left(O;n_{Ox},n_{Oy},n_{O}\right)$, as shown in Figure 1, the position vectors expand as [35]
(11)$$r_{0j}\left(x,y\right)=r_{x}x+r_{y}y+\frac{1}{2}\left(\frac{\partial^{2}w}{\partial x^{2}}x^{2}+2\frac{\partial^{2}w}{\partial x\partial y}xy+\frac{\partial^{2}w}{\partial y^{2}}y^{2}\right)n_{O}+\cdots$$

According to the geometric presumption (2), we ignore the influence of the in-plane deformations on the out-of-plane deformations. Therefore, the deformed position vector of the point j(x,y) is simplified as
(12)$$r_{Oj}=xn_{Ox}+yn_{Oy}+\frac{1}{2}\left(\frac{\partial^{2}w}{\partial x^{2}}x^{2}+2\frac{\partial^{2}w}{\partial x\partial y}xy+\frac{\partial^{2}w}{\partial y^{2}}y^{2}\right)n_{O}$$

According to Equation (12), the unit vectors in Equation (7) are
(13)$$\begin{array}{l} n_{O}=\left(0,0,1\right),\;d_{O3}=\frac{1}{l_{O3}}\left(1,0,\frac{l_{0}}{2}\frac{\partial^{2}w}{\partial x^{2}}\right),\\ d_{O1}=\frac{1}{l_{O1}}\left(-\frac{1}{2},\frac{\sqrt{3}}{2},\frac{l_{0}}{2}\left(\frac{1}{4}\frac{\partial^{2}w}{\partial x^{2}}-\frac{\sqrt{3}}{2}\frac{\partial^{2}w}{\partial x\partial y}+\frac{3}{4}\frac{\partial^{2}w}{\partial y^{2}}\right)\right),\\ d_{O2}=\frac{1}{l_{O2}}\left(-\frac{1}{2},-\frac{\sqrt{3}}{2},\frac{l_{0}}{2}\left(\frac{1}{4}\frac{\partial^{2}w}{\partial x^{2}}+\frac{\sqrt{3}}{2}\frac{\partial^{2}w}{\partial x\partial y}+\frac{3}{4}\frac{\partial^{2}w}{\partial y^{2}}\right)\right) \end{array}$$

Here,
(14)$$\begin{array}{l} l_{O1}=\sqrt{1+\frac{l_{0}^{2}}{4}{\left(\frac{1}{4}\frac{\partial^{2}w}{\partial x^{2}}-\frac{\sqrt{3}}{2}\frac{\partial^{2}w}{\partial x\partial y}+\frac{3}{4}\frac{\partial^{2}w}{\partial y^{2}}\right)}^{2}},\\ l_{O2}=\sqrt{1+\frac{l_{0}^{2}}{4}{\left(\frac{1}{4}\frac{\partial^{2}w}{\partial x^{2}}+\frac{\sqrt{3}}{2}\frac{\partial^{2}w}{\partial x\partial y}+\frac{3}{4}\frac{\partial^{2}w}{\partial y^{2}}\right)}^{2}},\\ l_{O3}=\sqrt{1+\frac{l_{0}^{2}}{4}{\left(\frac{\partial^{2}w}{\partial x^{2}}\right)}^{2}} \end{array}$$

Substituting Equations (13) and (14) into inversion angle terms in Equation (7), and through complicated but straightforward calculations, gets
(15)$$\left|n_{O}\times d_{O1}\right|=\left|n_{O}\times d_{O2}\right|=\left|n_{O}\times d_{O3}\right|=1-\frac{9r_{0}^{2}}{32}{\left(2H\right)}^{2}$$

Considering [11,19]
(16)$$\sum_{i=1}^{3}\left(1-n_{O}\cdot n_{i}\right)=\frac{3r_{0}^{2}}{4}\left[{\left(2H\right)}^{2}-2K\right]$$
and substituting Equations (10), (15) and (16) into Equation (7), we get the continuity potential energy density F as
(17)$$F=\frac{U}{A_{0}}=\frac{1}{2}k_{B}{\left(2H\right)}^{2}+k_{G}K+\frac{1}{2}k_{b}{\left(2J\right)}^{2}-k_{g}Q$$

Here, $A_{0}=3\sqrt{3}l_{0}^{2}/4$ is the area occupied by an atom. The stiffness coefficients in Equation (17) are
(18)$$\begin{array}{l} k_{b}=\left(\frac{9k_{l}l_{0}^{2}}{16}+3k_{\theta }\right),\;k_{g}=\left(\frac{3k_{l}l_{0}^{2}}{8}+4k_{\theta }\right),\\ k_{B}=\frac{\left(9k_{\chi }+24k_{\omega }\right)l_{0}^{2}}{16},\;k_{G}=-\frac{3k_{\omega }l_{0}^{2}}{2} \end{array}$$

Through an analogy with the classical Föppl-von Karman plate theory, the Poisson’s ratio ν and the 2D Young’s modulus Y can written as
(19)$$\nu =1-k_{g}k_{b}^{-1},\;Y=k_{b}\left(1-\nu^{2}\right)$$

Equation (17) shows that the continuous limit deformation energy of h-BNs has the same format as the classical Föppl-von Karman plate theory. Our new theory can also be used to model the mechanical properties of graphene if the graphene’s force field parameters are used. Since different researchers use different force field parameters, we calculate the mechanical parameters of h-BN and graphene for the force field parameters of Table 1, with the results as shown in Table 2. In order to compare the present results and the existing ones, Table 2 also lists the typical values obtained by DFTB [1], the bond orbital theory (BOT) [11], CM [19,20,23], and Experiments [36,37]. By comparing our theoretical results and pre-exciting studies, it is found that the in-plane elastic parameters of graphene and h-BN obtained by our approach are in good agreement with the pre-exciting results. However, the out-of-plane bending stiffness and torsion stiffness are different with different force field parameters. This result reveals a key connection: the out-of-plane mechanical properties significantly depend on the atomic force field. For example, the torsional stiffness comes from the dihedral angle terms. Therefore, if the dihedral angle terms are omitted in the MD calculations, the torsional properties of graphene and h-BN will be missed. In Table 2, “Present with [*]” means the values are calculated by Equations (18) and (19) with the force field parameters in Ref. [*]; DFTB means the density functional tight-binding calculations; BOT means the bond orbital theory; and CM means continuous model.

## 3. Discussion

Equation (17) indicates that the h-BN’s deformation energy density is consistent with the classical Föppl-von Karman plate theory. However, Equation (18) displays that the h-NB’s in-plane parameters, $k_{b}$ and $k_{g}$, are independent of the out-of-plane parameters, $k_{B}$ and $k_{G}$. This independence stems from the different atomic mechanisms between the in-plane stiffness and the out-of-plane stiffness. The changes in bond length and bond bending angle allow the ability to resist the in-plane tension and shear. The changes in the inversion angles and dihedral angles provide the ability to resist the out-of-plane bending. Furthermore, the anti-twisted ability only comes from the changes in the dihedral angles. In the macroscopic plates and shells, the in-plane stiffness comes from the potential energy increase induced by the midplane stretching and shear, while the out-of-plane stiffness comes from the opposite stretching states in the two sides of the midplane [16]. This makes the macroscopic plate’s in-plane and out-of-plane stiffness meet with Equation (3). However, From Table 2 and Equation (18), gets
(20)$$\frac{k_{B}}{k_{b}}=\frac{3k_{\chi }+8k_{\omega }l_{0}^{2}}{3k_{l}l_{0}^{2}+16k_{\theta }}\neq \frac{-k_{G}}{k_{g}}=\frac{12k_{\omega }l_{0}^{2}}{3k_{l}l_{0}^{2}+32k_{\theta }}$$

Equation (20) shows that the h-BN does not meet Equation (3). Therefore, it is impossible to relate the in-plane and out-of-plane mechanical parameters through the equivalent thickness. This indicates that the Yakobson paradox comes from the independence between the in-plane stiffness and the out-of-plane stiffness. From the bond orbital theory (BOT) based on the TB theory, the characteristics of the h-BN’s chemical bond are similar to graphene if the polarity of B and N atoms is ignored [11,25]. Therefore, it can be inferred from the graphene’s BOT that the h-BN’s in-plane stiffness may come from the bond stretching and bending, while the out-of-plane stiffness may come from the changes in the π-bonds.

Up to now, researchers have widely accepted that the 2D Young’s modulus and the Poisson’s ratio can correctly describe the h-BN’s in-plane mechanical properties under finite deformations. Nevertheless, how to describe the out-of-plane mechanical properties is still a controversial open question. Here we will use the present theory to clarify some arguments. Following the classic plate theory, we introduce three auxiliary functions as $\kappa_{xx}=-\partial^{2}w/\partial x^{2}$, $\kappa_{yy}=-\partial^{2}w/\partial y^{2}$ and $\kappa_{xy}=-\partial^{2}w/\partial x\partial y$ (known as bending curvatures [15,17]), so two bending moments and one twisting moment are
(21)$$\begin{array}{l} M_{xx}=\frac{\partial F}{\partial \kappa_{xx}}=k_{B}\kappa_{xx}+\left(k_{B}+k_{G}\right)\kappa_{yy},\\ M_{yy}=\frac{\partial F}{\partial \kappa_{yy}}=\left(k_{B}+k_{G}\right)\kappa_{xx}+k_{B}\kappa_{yy},\\ M_{xy}=\frac{1}{2}\frac{\partial F}{\partial \kappa_{xy}}=k_{G}\kappa_{xy} \end{array}$$

Equations (18) and (21) reveal: (1) the Gaussian stiffness, $k_{G}$, describes the h-BN’s ability to resist twisting, and this ability only comes from the dihedral angle term of the force field; (2) the bending stiffness $k_{B}$ comes from both the dihedral angle and inversion angle terms; (3) the Gaussian stiffness influences on two bending moments. There are two different options for the out-of-plane terms of the used force fields. The first option considers only the inversion angle terms, and this option makes $k_{G}=0$, $M_{xx}-M_{yy}=0$. It can be found from Equation (2) that this situation is impossible in classical plate theory because $k_{G}=0$ means |ν|=∞. Moreover, this situation also leads to two confusing results: two bending moments are always equal, and the equivalent thickness of the h-BN depends on the type of loads [17,22]. The other option considers only the dihedral angle terms and can get $k_{B}=-k_{G}$ from Equation (18). This means that the bending stiffness is identical to torsional stiffness. Furthermore $M_{xx}=k_{B}\kappa_{xx}$ and $M_{yy}=k_{B}\kappa_{yy}$, namely, a bending moment is only related to a bending curvature. Comparing Equations (21) and (2), one can find that $k_{B}=-k_{G}$ corresponds to the classical plate with Poisson’s ratio ν=0. According to the differential geometry theory, the Gauss-Bonnet integral formula has [35]: $\iint_{S}KdS=2\pi -\sum_{i=1}^{n}a_{i}-\oint_{C}k_{g}ds$, here $k_{g}$ is the geodesic curvature of the boundary curve, and $a_{i}$ is the exterior angle at the corner point. If the boundaries of an h-BN sheet are immovable, then $\oint_{C}k_{g}ds=0$ and $2\pi -\sum_{i=1}^{n}a_{i}=0$, further gets $\iint_{S}KdS=0$. For this case, the Gaussian curvature K does not appear in the balance equations and their boundary conditions, which it does not affect the deformation results of the structures. However, if the structures have free boundaries, the Gaussian curvature appears in the boundary conditions and affects the deformations [15,16]. Therefore, calculations on an h-BN shell with free boundaries may make mistakes if the force field’s four-body terms only include the inversion term or dihedral angle term. In addition, from the geometric point of view, the Gaussian curvature K=0 means that the surface is developable. If a continuous plate theory of h-BNs does not include Gauss curvature, the theory only describes special cases in which an h-BN is curled as a developable surface.

The above discussions are based on our analytical theory. Compared with the DFT calculations, the main advantages of our new theory are as follows. First, the new theory demonstrates that the modified Föppl-von Karman plate theory can describe h-BN’s mechanical properties. Second, the new theory provides analytical formulas of h-BN’s mechanical parameters with the DREIDING force field. DFT calculations cannot give these relationships. Third, this theory discovers that the bending and torsion stiffnesses do not depend on the h-BN’s thickness. This gives a clear physical explanation of the Yakobson paradox revealed by the DFT calculations. In fact, the Yakobson paradox exists in h-BNs and graphene, and other 2D nanomaterials [1,38,39], for example the single layer $MoS_{2}$.

Recently, heterogeneous structures composed of the h-BN and other 2D nanomaterials have been widely considered [40,41]. The nonbonding van der Waals interaction is essential in heterogeneous structures. The van der Waals interaction between the nano curved surface and the in-surface or out-surface particles can be expressed as a unified function of the mean curvature and Gaussian curvature of the curved surface [42,43]. How to consider the van der Waals interaction in the present theory needs further research.

## 4. Conclusions

In this research, through the continuity of the DREIGING force field, we prove that the h-BN’s mechanical properties can model as a Föppl-von Karman plate theory with independent in-plane and out-of-plane mechanical parameters under finite deformations. The in-plane stiffness depends on the bond stretching and bond bending; the out-of-plane bending stiffness depends on the changes in the inversion angles and the dihedral angles; the torsional stiffness (the Gaussian stiffness) only depends on the changes in the dihedral angles. Since the force field’s sources of the in-plane and out-of-plane stiffness are independent, it is unnecessary to relate the in-plane and the out-of-plane mechanical parameters through the equivalent thickness of h-BNs. Our present theory provides the Yakobson paradox with an atomic explanation of the one-atom-thick 2D nanomaterials. This research is heuristic in modelling other 2D nanomaterials through the classical macroscopic elastic theory.

## Figures and Tables

**Figure 1 nanomaterials-11-03113-f001:**
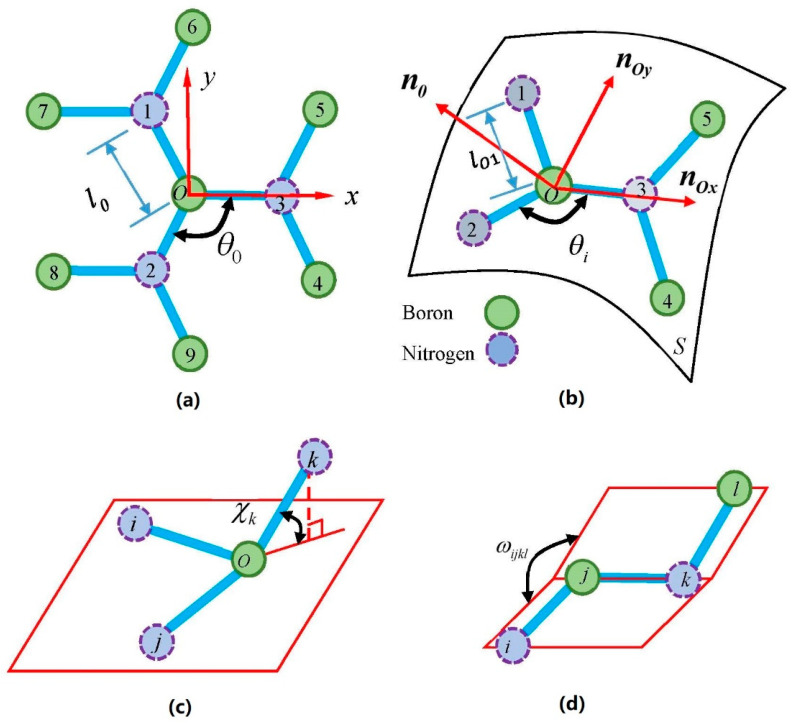
Fragment of single-layer h-BN: (**a**) undeformed structure; (**b**) atoms in the deformed surface; (**c**) four-body inversion angle; (**d**) four-body dihedral angle.

**Table 1 nanomaterials-11-03113-t001:** The force field parameters of h-BN and graphene.

	$k_{l}\left(eV/nm^{2}\right)$	$k_{\theta }\left(\mathrm{eV}/\mathrm{rad}\right)$	$k_{\chi }\left(\mathrm{eV}/\mathrm{rad}\right)$	$k_{\omega }\left(\mathrm{eV}/\mathrm{rad}\right)$	Source
h-BN	3279	4.34	1.73	0.22	[26]
Graphene	4068	5.46	0	1.73	[30]
Graphene	4554	4.34	1.73	1.08	[26]
h-BN	3036	4.34	0	3.90	[31]

**Table 2 nanomaterials-11-03113-t002:** The mechanical parameters of h-BN and graphene.

	$k_{b}\left(\mathrm{eV}/nm^{2}\right)$	$k_{g}\left(\mathrm{eV}/nm^{2}\right)$	Y(N/m)	ν	$k_{B}\left(\mathrm{eV}\right)$	$k_{G}\left(\mathrm{eV}\right)$	Source
Graphene	2531	2153	345	0.149	1.44	−1.52	DFTB [1]
Graphene	2492	1795	368	0.28	2.28	−1.59	BOT [11]
Graphene	2488	1687	352	0.34	1.17	−0.75	CM [19]
Graphene	--	--	--	--	1.40	−1.63	CM [20]
Graphene	--	--	340	--	--	--	Experiments [36]
h-BN	1770	1397	271	0.211	1.34	--	DFTB [1]
h-BN	1812	1468	265.9	0.29	--	--	ANA [4]
h-BN	2404	1731	355	0.28	2.1	−1.46	BOT [11]
h-BN	1469	1146	224	0.22	2.13	--	CM [23]
h-BN	--	--	289	--	--	--	Experiments [37]
Graphene	2468	1977	380	0.20	1.99	−1.25	Present with [26]
Graphene	2386	2008	386	0.15	2.00	−2.00	Present with [30]
h-BN	1897	1583	295	0.17	1.00	−0.25	Present with [26]
h-BN	1792	1513	280	0.16	4.51	−4.51	Present with [31]

## Data Availability

Not applicable.

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
