# Peer review of "An Atomistic-Based Nonlinear Plate Theory for Hexagonal Boron Nitride"

_nanomaterials, 2021, doi:10.3390/nano11113113_

Round 1
Reviewer 1 Report
In this manuscript the authors have proposed an Atomistic-based Nonlinear Plate Theory for Hexagonal Boron Nitride. In general the manuscript presents interesting results, in addition it is clearly in written. However, of my point of view, some question should be explained.
1)How is compared the results here founded as that obtained through other methods. For example, first principles method.
2) What advantage of this method when compared to DFT method, for example ?
In summary, I recommend the manuscript after of these explanations .
Author Response
Major comments: In this manuscript the authors have proposed an Atomistic-based Nonlinear Plate Theory for Hexagonal Boron Nitride. In general the manuscript presents interesting results, in addition it is clearly in written. However, of my point of view, some question should be explained.
1) How is compared the results here founded as that obtained through other methods. For example, first principles method.
2) What advantage of this method when compared to DFT method, for example ?
In summary, I recommend the manuscript after of these explanations .
Reply: We thank the reviewers for the comments.
(1) To check the reliability of our theory, we list the theoretical results obtained by our analytical formulas with two different sets of the DREIDING force field parameters in Table 2. Furthermore, we also list several sets of widely cited results obtained by first-principles calculations, continuity theories, and experiments in Table 2. By comparing our theoretical results and pre-exciting studies, it is found that the in-plane elastic parameters of graphene and h-BN obtained by our approach are in good agreement with the pre-exciting results. However, the out-of-plane bending stiffness and torsion stiffness are different with different force field parameters. This paper revealed a key connection: the out-of-plane mechanical properties significantly depend on the atomic force field. For example, the torsional stiffness comes from the dihedral angle terms. Therefore, if the dihedral angle terms are omitted in the MD calculations, the torsional properties of graphene and h-BN will be missed. In other words, to accurately describe the mechanical properties of graphene and h-BN, the bond length, bond angle, off-plane angle, and dihedral angle are all necessary. In addition, we have added the two results from experiments in Table 2 (Ref. [36] and [37]) and supplemented the discussion on page 8.
(2) Compared with the DFT method, the main advantages of our theory are as follows. First, it is the first theoretical proof that the modified Föppl-von Karman plate theory can describe h-BN’s mechanical properties. Second, our theory gives analytical formulas of h-BN’s mechanical parameters with the DREIDING force field. DFT calculations cannot give these relationships. Third, our theory discovers that the bending stiffness and torsion stiffness do not depend on the h-BN’s thickness. This gives a clear physical explanation of the Yakobson paradox revealed by the DFT calculations. We have supplemented this issue on page 11 of the revision.
The above modifications are written in red.
Reviewer 2 Report
Authors have proposed an atomistic based plate theory for modelling h-BN structures. The results are interesting. Some suggestions to further improve the manuscript before it can be accepted in present form: 1. In addressing the thickness parameter of h-BN structures, there are already some pre-existing studies which need to be included in literature reference. 2. In Table 2, authors need to explain more clearly on how the parameters were obtained using different reference literatures. 3. What are the significant advantage of using the proposed modelling parameters over the existing potential functions. For instance, Tersoff with MD offers faster computation for a large scale of atoms. 4. It would also be useful to compare the findings with some experimental studies.Author Response
Reviewer's Comments: Authors have proposed an atomistic based plate theory for modelling h-BN structures. The results are interesting. Some suggestions to further improve the manuscript before it can be accepted in present form: 1. In addressing the thickness parameter of h-BN structures, there are already some pre-existing studies which need to be included in literature reference. 2. In Table 2, authors need to explain more clearly on how the parameters were obtained using different reference literatures. 3. What are the significant advantage of using the proposed modelling parameters over the existing potential functions. For instance, Tersoff with MD offers faster computation for a large scale of atoms. 4. It would also be useful to compare the findings with some experimental studies.
Reply: We thank the reviewers for the comments.
(1) The h-BN’s thickness is related to its bending stiffness and torsional stiffness. However, the Yakobson paradox induced by the h-BN’s thickness has received less attention compared with graphene. In order to facilitate readers to understand this issue, we add two review references (References [38] and [39]) and add comments on page 11 of the paper.
(2) We have added the methods of obtaining mechanical parameters in the references in Table 2.
(3) The present paper has two primary purposes. First, the paper proves theoretically that the modified classical nonlinear plate theory can describe the h-BN’s elastic behavior. Second, the paper obtains the relationships between mechanical parameters and the DREIDING force field. The relationships between mechanical parameters and force field parameters allow us to explain the physical source of the Yakobson paradox. Our theory also explains clearly the independence between out-of-plane stiffness and thickness. So parameters’ obtaining of the potential functions is not our key point. Nevertheless, we believe this question is essential, and we will research it in another paper. We add comments on page 11 of the paper.
(4) In Table 2, we have added two experimental references of graphene and h-BN (Ref. [36] and [37]) and commented on the two references on page 8 of the paper.
The above modifications are written in red.
Round 2
Reviewer 1 Report
The authors answered all my questions, therefore the manuscript can be accepted .